Increased contribution of parasites in microbial eukaryotic communities of different Aegean Sea coastal systems

Meziti Alexandra a.meziti@aegean.gr 1
Smeti Evangelia 1 2
Daniilides Daniil 3
Spatharis Sofie 4 5
Tsirtsis George 1
Kormas Konstantinos A. 6
1 Department of Marine Sciences, University of the Aegean , Mytilene , Greece
2 Institute of Marine Biological Resources & Inland Waters, Hellenic Centre for Marine Research , Anavissos , Greece
3 Faculty of Biology, Department of Ecology and Systematics, University of Athens , Athens , Greece
4 School of Life Sciences, University of Glasgow , Glasgow , United Kingdom
5 Institute of Biodiversity, Animal Health and Comparative Medicine, University of Glasgow , Glasgow , United Kingdom
6 Department of Ichthyology & Aquatic Environment, University of Thessaly , Volos , Greece
Venmathi Maran Balu Alagar
Electronic publication date: 2023 Dec 19
Publication date: 2023
Volume: 11
Electronic Location ID: e16655
Received 2023 Aug 10; Accepted 2023 Nov 21
Copyright: ©2023 Meziti et al.
Copyright year: 2023
Copyright holder: Meziti et al.
License: This is an open access article distributed under the terms of the Creative Commons Attribution License, which permits unrestricted use, distribution, reproduction and adaptation in any medium and for any purpose provided that it is properly attributed. For attribution, the original author(s), title, publication source (PeerJ) and either DOI or URL of the article must be cited.
License URL: https://creativecommons.org/licenses/by/4.0/

Keywords: Coastal systems, Protist communities, Parasites, 18S rRNA

Funding: ‘ECOGENE: The relative role of niche and neutral mechanisms in controlling phytoplankton genetic and morphological diversity’ (Code Number 4691) ARISTEIA II of the Operational Program ‘Education and Lifelong Learning’ (Action’s Beneficiary: General Secretariat for Research and Technology) The European Social Fund (ESF) and the Greek State This research was conducted within the research project ‘ECOGENE: The relative role of niche and neutral mechanisms in controlling phytoplankton genetic and morphological diversity’ (Code Number 4691), implemented within the framework of the Action ARISTEIA II of the Operational Program ‘Education and Lifelong Learning’ (Action’s Beneficiary: General Secretariat for Research and Technology) and by the European Social Fund (ESF) and the Greek State. The funders had no role in study design, data collection and analysis, decision to publish, or preparation of the manuscript.

==============================
Background-Aim

Protistan communities have a major contribution to biochemical processes and food webs in coastal ecosystems. However, related studies are scarce and usually limited in specific groups and/or sites. The present study examined the spatial structure of the entire protistan community in seven different gulfs and three different depths in a regional Mediterranean Sea, aiming to define taxa that are important for differences detected in the marine microbial network across the different gulfs studied as well as their trophic interactions.

Methods

Protistan community structure analysis was based on the diversity of the V2–V3 hypervariable region of the 18S rRNA gene. Operational taxonomic units (OTUs) were identified using a 97% sequence identity threshold and were characterized based on their taxonomy, trophic role, abundance and niche specialization level. The differentially abundant, between gulfs, OTUs were considered for all depths and interactions amongst them were calculated, with statistic and network analysis.

Results

It was shown that Dinophyceae, Bacillariophyta and Syndiniales were the most abundant groups, prevalent in all sites and depths. Gulfs separation was more striking at surface corroborating with changes in environmental factors, while it was less pronounced in higher depths. The study of differentially abundant, between gulfs, OTUs revealed that the strongest biotic interactions in all depths occurred between parasite species (mainly Syndiniales) and other trophic groups. Most of these species were generalists but not abundant highlighting the importance of rare species in protistan community assemblage.

Conclusion

Overall this study revealed the emergence of parasites as important contributors in protistan network regulation regardless of depth.

Introduction

Single-celled eukaryotes play an important role in numerous essential ecological and biogeochemical processes within marine ecosystems globally, serving as primary producers, predators, decomposers and parasites (Sherr & Sherr, 2002). Despite their importance, the distribution of protists within the pelagic environment as well as the role of protistan interactions within marine food webs remains rather unclear. The emergence of high-throughput sequencing (HTS) in the last decades has enabled more in depth studies on the distribution, metabolic activity, trophic status and biological interactions of marine protists, mainly focusing on over depth and/or seasonal changes (Giner et al., 2020; Ollison et al., 2021; Yeh & Fuhrman, 2022). Most of the published literature is focusing on depth and/or seasonal changes in oceanic samples reaching depths of up to 4,000 m (Giner et al., 2020). Other studies also working in open ocean sites have tested protists dispersal in ocean surface showing that only 0.35% of the operational taxonomic units (OTUs) detected are indeed cosmopolitan implying dispersal limitations (Burki, Sandin & Jamy, 2021; De Vargas et al., 2015) mostly set by ocean currents for large heterotrophs and by environmental conditions for small-bodied phototrophs (Sommeria-Klein et al., 2021).

Although the importance of unicellular eukaryotes, has been recognized as highly relevant to several Marine Strategy Framework Directive descriptors, notably D1 (Biological Diversity), D4 (Food webs) and D5 (Eutrophication) (Caruso et al., 2016) such studies in coastal systems are limited (Genitsaris et al., 2015; Massana et al., 2015; Skouroliakou et al., 2022), mostly focusing in temporal changes, while over space and depth studies are lacking. Understanding the drivers behind protistan distribution and interactions in coastal ecosystems is crucial since these systems tend to be very different from oceanic environments, being much more influenced by terrestrial inputs, adhering to very different hydrodynamic processes and being subject to very different light regimes.

In order to answer questions related to protist variation over space and depth (related to light penetration) and to highlight the trophic interactions driving these changes, we studied protistan community composition across different coastal sites and functional depths (surface, depth of Secchi disc, and 2× depth of Secchi disc), from seven different gulfs in the Aegean Sea (regional Mediterranean Sea), using 18S rRNA sequencing. Studies on protistan community composition on the specific area are scarce and limited and are focused on specific protistan groups separately; i.e., ciliates (Giannakourou et al., 2014; Meziti & Kormas, 2022). Studies using novel metabarcoding techniques investigating the whole community are scarce and are focused in specific sites; i.e., Thermaikos Gulf (Genitsaris et al., 2020; Genitsaris et al., 2022) or groups; i.e., phytoplankton (Spatharis et al., 2019).

In this study we tried to define taxa that are important for differences detected in the marine microbial network across the different gulfs studied as well as their characteristics in terms of trophic level (autotrophs, mixotrophs, heterotrophs, parasites), niche breadth (generalists, specialists) and abundance (abundant, rare). Our target was to elucidate whether these taxa and their trophic and ecological characteristics differentiate between different depths, as well as their associations. Although we expected that phytoplankton species along with their grazers would have key role in the upper layers, and that parasites would be more important in higher depths, it was shown that the latter were important in all depths across different gulfs exhibiting significant interactions with all trophic groups and representing key species for the explanation of spatial variation.

Materials & Methods

Collection of samples and DNA extraction

Samples were collected from seven gulfs in the Aegean Sea, Greece, set within a polygonal area of 72,600 km2 (Table 1; Fig. S1), characterized by a range of environmental conditions due to differences in hydrology, geomorphology, substrate, terrestrial runoff and local anthropogenic pressures (Spatharis et al., 2019). Within each site, five stations were sampled at 1 m (surface), at the Secchi disc depth (Secchi) and at the two times Secchi disc depth (2xSecchi) or maximum depth, depending on site depth (Table S1), collecting 1 l seawater with a Niskin type sampler. In some cases duplicates or triplicates were collected in order to check sampling and sequencing consistency (Table S1).

Table 1 Characteristics of the seven coastal sites in the Aegean Sea including the distance from coast, Secchi depth and max depth at the five stations (median; minimum and maximum values in brackets).

Site code	Site name	Distance of stations from coast (km)	Secci depth (m)	Maximum depth (m)	
S	Saronikos Gulf	1.03 (0.24–1.87)	16 (13–30)	19 (15–29)	
T	Thermaikos Gulf	2.78 (2.18–3.41)	10 (8–25)	24 (18–27)	
KV	Kavala Gulf	4.26 (3.70–9.56)	30 (26–34)	35 (30–45)	
LK	Kodias Bay	0.24 (0.09–0.44)	16 (12–23)	16 (13–24)	
M	Moudros Gulf	0.60 (0.46–1.87)	10 (10–17)	11 (10–17)	
G	Gera Gulf	0.72 (0.3-2.45)	10 (916)	11 (1-17)	
K	Kalloni Gulf	1.57 (1.16–3.29)	7 (6–8)	13 (8–16)	

Sampling took place during July 2014 and was carried out within 19 days (5–24 July) to minimize the effect of temporal variation on assemblage composition. July was selected for sampling as previous studies have shown that physico-chemical variables and phytoplankton composition are relatively stable during this summer period, at least for a subset of the coastal areas included in the present study (Spatharis et al., 2007; Naselli-Flores et al., 2003); this is in contrast with winter months, during which episodic rainfall events, and strong wind mixing of the water column and sediment resuspension—which at this large scale may not simultaneously affect all sites—add noise that could distort food web dynamics. Water samples were filtered using low vacuum filtration (<150 mmHg) on 0.2 µm isopore filters (Sartorius Stedim Biotech, Germany).

Environmental parameters

At each station and depth, several environmental parameters were recorded. Specifically, salinity and temperature were recorded onsite, while 3 l seawater samples were collected with a Niskin type sampler for later nutrient measurement (nitrate, nitrite phosphate, silicate, organic nitrogen, organic phosphorus). Organic nitrogen and phosphorus were strongly correlated with dissolved inorganic nitrogen (DIN) and dissolved inorganic phosphorus (DIP) and were therefore excluded from further analysis. The covariates used in the analysis were thus salinity, temperature, DIN, DIP and silicate.

DNA extraction and sequencing

DNA extraction from filters was performed with the MoBio Power Soil kit (MoBio Inc. Carlsbad, CA, USA) following its standard protocol with minor modifications for filters processing.

Sequencing of the V2–V3 region of the 18S rRNA gene was performed upon amplification using the primer pair 18S-82F (5′-GAAACTGCGAATGGCTC-3′) (López-García et al., 2003) and Euk-516r (5′-ACCAGACTTGCCCTCC-3′) for Eukaryotes (Amann et al., 1990). Construction of libraries was performed by ‘Genes Diffusion’ company (Lille, France) and amplicons were finally sequenced with Illumina MiSeq PE 2x300 (CNRS-UMR8199; Illumina, Lille, France).

18S rRNA gene amplicon analysis

Processing of the resulting sequences, i.e., sequence assembly and quality control, was performed with the MOTHUR software (v 1.35) (Schloss et al., 2009). Only sequences with ≥480 bp, no ambiguous bases and homopolymers shorter than 8 bp were considered for further analysis. These sequences were aligned using the SILVA SSU database (release 119) (Pruesse et al., 2007). Chimeras were removed using the Uchime Software (Edgar et al., 2011). All sequences were binned into operational taxonomic units (OTUs) and were clustered (average neighbour algorithm) at 97% sequence similarity. Single singletons, that appeared only once in the whole dataset, were removed using MOTHUR (v 1.35). Coverage values were calculated with MOTHUR (v 1.35) as well as diversity indices. The batch of sequences from this study has been submitted in NCBI Short Read Archive under accession number PRJNA515026. Taxonomic classification was assigned using BLAST (Altschul et al., 1990) on the Protist Ribosomal Reference (PR2 version 4.14.0) curated Database (built on GenBank; June 2021), containing 197,602 sequences (Guillou et al., 2013).

Similarity profiles, multivariate analysis and differentially abundant features

The Bray–Curtis similarities coefficients were calculated, based on non-transformed number of reads in order to identify relationships between the samples, using R (R Core Team, 2020). NMDS ordination plots were prepared in R (package vegan; Oksanen et al., 2020) using Bray-curtis similarities.

Differentially abundant OTUs between gulfs, were identified with DESeq2 package version 3.0.2 (Anders & Huber, 2010; Love, Huber & Anders, 2014) which performs differential analysis of count data, estimating dispersions and fold changes, thus enabling a more quantitative analysis focused on the strength rather than the mere presence of differential expression.

Abundant and rare OTUs, generalists and specialists and trophic groups

OTUs were classified as abundant or rare based on total relative abundances of on non-transformed number of reads and on the thresholds used in previous studies (Logares et al., 2014) using >1% for abundant OTUs and <0.2% for rare OTUs. Habitat specialization was calculated as described in Székely & Langenheder (2014) using Levins’ niche width index (B) (Levins, 1968), where B=1/∑i=1Npij2 (pij: the proportion of OTU j in sample I; N: the total number of samples). Thus B, although is not taking the environmental conditions in a local community into account, is describing the extent of niche specialization based on the distribution of OTUs abundances.

Specialization categories were set along arbitrary cut-off values (B > 15, B:10–15, B:1–10 B = 1) with B = 1 corresponding to extreme specialists (one sample) and B > 15 corresponding to top generalists (>66% of available habitats). Niche width was calculated based on the presence of each OTU at all sites and depths (112 samples) as each sample was considered to be a different habitat in terms of physico-chemical conditions. OTUs were sorted into major trophic groups, such as autotrophs, nanoheterotrophs (picograzers), nanoplankton grazers (nanograzers), microplankton grazers (micrograzers), mixotrophs and parasites following the classification shown in Genitsaris et al. (2016).

Correlation and network analysis

For the network analysis, we focused on the OTUs indicated by DeSeq as they were considered ‘key’ species for community structure regulation. The relationships/interactions amongst these OTUs were characterized by computing the maximal information coefficient (MIC) between each OTU pair (Reshef et al., 2016), calculated using MICtools v1.1.4 (Albanese et al., 2018). MIC captures associations between data and provides a score that represents the strength of the relationship between data pairs. MIC values > 0.5, corresponding to a p-value < 0.05, were used for the networks. For networks visualization Cytoscape 3.0 (Smoot et al., 2011) was used.

Results

Protist community structure

Overall 4,389,394 sequences were obtained resulting in 5,005 operational taxonomic units (OTUs) in all 112 samples. Diversity indices indicated that the highest richness as well as Shannon diversity were observed in LK, while in S the respective values were the lowest ones followed by T and KV (Table S2). Shannon diversity index was stable over depth in most gulfs apart from S, T and KV where diversity increased in deeper layers (Table S2).

Protistan community composition (PCC) similarities within gulfs decreased over depth apart from K, S and KV (Fig. S2 a). Regarding between gulf similarities anova results on hellinger transformed OTUs abundances indicated several significant differences (p < 0.05; Table S3A) at the surface level while at Secchi depth the only difference was observed between T and K (p = 0.041) and at 2xSecchi no differences were observed (Table S3A). Bray-Curtis similarities between gulfs ranged from 0.25 to 0.30 with the highest values being observed in surface (Fig. S2B). This range was very low, although differences were significant between surface and the other depths (t-test Monte-Carlo permutation; p = 0.0001 for both Secchi and 2xSecchi). Overall these results implied more distinct between gulfs, but homogenous within gulfs, communities at the surface level while communities between gulfs were less distinguishable at higher depths.

Similar results were observed after analysis with NMDS showing spatial grouping, mostly for the surface and Secchi level samples (Fig. 1). Ordination stress was quite high (0.12, 0.14 and 0.18 for different depths respectively). The only environmental parameters that were important for these ordinations (p < 0.001) were silicate and ammonia (p = 0.00099) for both surface and secchi samples, NO2 only for surface (p = 0.00099) and phosphates and Chla (p = 0.00099) for Secchi depth. Finally at 2xsecchi depth only temperature appeared as significant for the ordination (Fig. 1).

Figure 1 NMDS plot of taxonomic distributions based on non-transformed OTUs abundances (97% similarity).

Surface (stress = 12.23), Secchi (stress = 14.30) and 2xSecchi (stress = 18.13) depths. Arrows indicate significance (p < 0.001) for the plotting of the samples.

Taxonomic classification for each gulf and depth revealed different patterns. Overall nine major taxonomic groups comprised ∼90% of taxonomic diversity in all samples (Fig. 2). Bacillariophyta, Dinophyceae and Syndiniales were the most abundant groups in most cases, exhibiting fluctuating abundances across gulfs and depths. However Chrysophyceae prevailed in S samples where Bacillariophyta decreased and Haptophyta exhibited their highest relative abundances in K (Fig. 2).

Figure 2 Average relative abundances of the nine major groups in all gulfs and depths.

Effect of depth in trophic groups; abundant OTUs and niche breadth

Overall autotrophic algae (shown in green colors in Fig. 2) abundance dropped from surface to deeper layers. It has to be noted though that in KV, M and LK samples 2xSecchi depth coincides with maximum depth; thus, these samples were collected from bottom water (Table S1). Overall parasites (mainly Syndiniales) relative abundances were constantly >10% in all gulfs (Fig. 2). Correlation analysis and the calculation of MIC scores for each depth separately, showed that in surface strong correlations were observed between all trophic groups apart from autotrophs, while in Secchi depth correlations were observed only between grazers and parasites and no correlations were observed in 2xSecchi (Table S4).

The whole dataset included 25 abundant (>1%) and 60 common (>0.2%) OTUs, accounting for 0.5% and 1.19% of the total OTUs number respectively (Table S5). However, in terms of relative abundances these OTUs accounted for >60% of protistan community composition. These abundant species belonged to all trophic groups, exhibiting different abundance patterns amongst gulfs and depths (i.e., prevalence of Chrysophyceae in S samples) (Fig. 3).

Figure 3 Taxonomic distributions, at the lowest level detected.

Taxonomic distributions, at the lowest level detected, of abundant OTUs (<1%) from surface (surf), Secchi(sec) and 2xSecchi depths (2xsec) samples. Scale corresponds to log transformed concentrations.

Amongst the 5,005 OTUs detected, 2,314 represented extreme specialists (present only in one site and depth), 2,038 represented specialists (present in <30% of all sites and depths; 1 < B < 10), 227 represented generalists (10 < B < 15) and 426 were top generalists (B > 15) reaching ∼50% of OTUs number/sample (Fig. S3A; Table S5). In most samples top generalists and generalists accounted for >90% of OTUs relative abundances (Fig. S3B), apart from secchi and 2xsecchi samples in KV, S and T and 2xSecchi in G. Overall the number of extreme specialists and specialists increased with depth with differences being significant between surface and 2xsecchi (p = 0.0126; F = 5.06).

The vast majority of abundant OTUs consisted of top generalists (17/25) and generalists (5/25), while a similar pattern was observed for common OTUs (50/60; 6/60) (Table S5). Only 17/426 top-generalists were abundant and 47/426 were common, while the rest were rare species. Similarly for generalists 216/227 species were rare (Table S5).

Differentially abundant OTUs across depths exhibit different trophic and niche breadth patterns

The number of differentially abundant OTUs across the seascape (using the binomial test and false discovery rate < 0.05), varied between depths, with 44, 23 and 51 OTUs at surface, Secchi, and 2XSecchi depths, respectively (Table S6). For surface samples these OTUs belonged to all trophic groups (Table S6) and more specifically included increased relative abundances of phototrophic algae in T, KV and G, of nanograzers in G and S, of mixotrophs in S and of parasites in KV and M (Fig. 4). For secchi depth, differences between sites were attributed to fluctuations of OTUs, belonging to mixotrophic and parasitic groups (Fig. 4) that both peaked in KV followed by S, while fluctuations of nanograzers were also important. Finally, for 2xsecchi, the majority of differences were attributed to OTUs that mostly belonged to heterotrophic and mixotrophic groups, although in specific sites such as LK, M, S and T increases of algae (Chrysophyceae) were responsible for gulf separation, while parasites peaked in KV (Fig. 4, Table S6). We have to highlight that in all depths >70% of differentially abundant OTUs belonged to rare OTUs (Table S6) showing the importance of rare members of the total protistan community for spatial differentiations in PCC.

Figure 4 Relative abundances of OTUs responsible for different gulf clusters categorized at different trophic groups.

Regarding these OTUs niche breadth, at surface 28/44 OTUs were top-generalists and generalists, while at Secchi and 2xSecchi depths the respective ratios were only 7/23 and 9/51 (Table S6), corroborating with the total number of specialists increasing with depth (Fig. S3A).

Amongst these generalists and top-generalists differentially abundant species only 4/28, 1/7 and 3/9 species were also abundant at surface, Secchi and 2Xsechhi, respectively (Table S6, Fig. 3). These OTUs were related to Leptocylindrus (OTU00031) and to Chaetoceros (OTU00036) in surface, to Chrysophyceae (OTU00022) in 2X Secchi, while the uncultured Syndiniales (OTU00056) was important for all depths and an uncultured Dinophyceae (OTU00004) was important in surface and 2XSecchi (Table 2). Apart from OTU00056, two more species, OTU00288 and OTU00737 were important for differences between gulfs in all depths. OTU00288 was a generalist although rare mixotrophic haptophyte, and OTU00737 was a rare specialist nanograzer belonging to Spirotrichea.

Table 2 OTUs identified as differentially abundant between gulfs, showing also increased MIC scores (>0.5) with other OTUs. Differential abundance detection depth (S: surface, S: Secchi depth, 2XS: 2xSecchi depth), niche breadth (tg: top generalist, g: generalist, s: specialist).

OTU	Sur	S	2XS	Abundance	Levin	Closest relative[GenbankID](% identity)	Taxonomic affiliation	Trophic group	
Otu00102	X		X	common	tg	Micromonas sp.[AY425318](96.68)	Cholorophyta	autotroph	
Otu00919	X			rare	tg	Nephroselmis pyriformis[AB058391](100)	Cholorophyta	autotroph	
Otu00022			X	abundant	g	Chrysophyceae Clade-H X sp [KC582999](99.79)	Chrysophyceae	autotroph	
Otu00064			X	rare	g	Paraphysomonas imperforata[AF109324](100)	Bacillariohyta	autotroph	
Otu00164	X			common	tg	MOCH-5 XXX sp [KC583020](100)	Bacillariohyta	autotroph	
Otu00713	X	X		rare	g	Chaetoceros affinis[MG972330](95.84)	Bacillariohyta	autotroph	
Otu01228		X		rare	s	Chaetoceros mitra[KX611427](97.04)	Bacillariohyta	autotroph	
Otu00119			X	common	tg	Leptocylindrus danicus[JX413558]100	Bacillariohyta	autotroph	
Otu00068	X			common	tg	Suessiales sp.[GQ483674](99.17)	Dinophyceae	mixotroph	
Otu00134	X	X		common	s	Yihiella yeosuensis[EF526743](99.15)	Dinophyceae	mixotroph	
Otu00265		X		rare	s	Dinophyceae sp.[JX188282](99.79)	Dinophyceae	mixotroph	
Otu00284	X	X		rare	tg	Dinophyceae sp.[KF130231](100)	Dinophyceae	mixotroph	
Otu00288	X	X	X	rare	g	Haptolina sp.[AB058358](100)	Haptophyta	mixotroph	
Otu00556		X		common	s	Dinophyceae sp.[KJ759720](96.27)	Dinophyceae	mixotroph	
Otu01101		X		rare	s	Dinophyceae sp.[DQ103863](94.58)	Dinophyceae	mixotroph	
Otu00124	X		X	abundant	s	Pelagostrobilidium minutum[FJ876959](99.37)	Ciliophora	nanograzer	
Otu00409	X	X		rare	tg	Strombidiidae K sp.[AF372790](98.75)	Ciliophora	nanograzer	
Otu00604	X		X	rare	s	Colpodea sp[EF023549](97.91)	Ciliophora	nanograzer	
Otu00737	X	X	X	rare	s	Strobilidiidae I sp.[KC488367](98.74)	Ciliophora	nanograzer	
Otu00824	X	X		rare	g	Eutintinnus medius[KT792925](100)	Ciliophora	nanograzer	
Otu00263	X			rare	s	Stephanoecidae Group D sp.[KJ763316](98.11)	Choanoflaggelida	nanohetrotroph	
Otu00386	X		X	rare	s	Choanoflagellida sp.[KJ759457](98.13)	Choanoflaggelida	nanohetrotroph	
Otu00472	X		X	rare	s	Acanthoecida sp.[KJ762979](98.29)	Choanoflaggelida	nanohetrotroph	
Otu00052		X	X	common	s	Blastodinium navicula[DQ317538](97.08)	Dinophyceae	parasite	
Otu00056	X	X	X	abundant	g	Dino-Group-I sp[AB252762](100)	Syndiniales	parasite	
Otu00084			X	common	g	Dino-Group-II sp[EU333053](99.13)	Syndiniales	parasite	
Otu00165	X	X		rare	s	Cryptomycotina sp.[AB191431](98.54)	Fungi	parasite	
Otu00177	X			rare	s	Dino-Group-II sp[EU793925](100)	Syndiniales	parasite	
Otu00281	X			rare	g	Dino-Group-II sp[KJ762603](96.86)	Syndiniales	parasite	
Otu00286			X	rare	s	Dino-Group-II sp[KC582953](96.38)	Syndiniales	parasite	
Otu00518			X	rare	s	Abeoformidae Group MAIP 2 X sp [JN832892](97.91)	Mesomycetazoa	parasite	
Otu00812	X	X		rare	g	Dino-Group-II sp[EU793924](95.42)	Syndiniales	parasite	
Otu00104	 	 	X	common	tg	Uncultured syndiniales clone PROSOPE.EM-5m.190[EU793933](98)	Syndiniales	parasite	

Correlations between differentially abundant OTUs; network analysis for different depths

Network analysis for the 44 surface species that were responsible for between gulfs differences indicated 155 significant relationships between 42 species while two species indicated no correlation (Table S7). Amongst these species 11 species exhibited 105 significant interactions amongst them as well as with 19 other species and were responsible for the majority of interactions observed (Fig. 5A,; Tables 2; S7).

Figure 5 Network diagram of MIC correlations (edges) between OTUs (nodes) responsible for the differences between gulfs.

(A) Surface, (B) Secchi and (C) 2xSecchi depths. The different colors represent different ecological categories, based on the major ecological role of each OTU in a marine ecosystem as determined by the literature (red: parasites, yellow: nanograzers/nanoheterotrophs, green: mixotrophs, blue: phototrophic algae). The weight of the edges is analogous to the strength of the connection (MIC value) and node size is analogous to node degree.

These eleven species belonged to algae (Chlorophyta, Bacillariophyta) that mostly interacted with parasites followed by other algae and mixotrophs, parasites (Syndiniales) apart from algae also interacted with mixotrophs, and mixotrophs (Haptophyta, Dinophyceae), nanoheterotrophs and nanograzers that interacted almost equally with all groups (Table S6). The highest MIC scores were observed between algae, mixotrophs and some parasites while the correlations with other heterotrophs were minor (Tables 2; S7). Most importantly these 11 OTUs belonged to generalists and top-generalists and only the nanonoheterotroph OTU00263 and the parasitic OTU00177 (Syndiniales) were specialists (Table S6). On the other hand, the Secchi network had much lower centrality (that is few central nodes that are close to all the other nodes) with only 15 species (out of 23) exhibiting 23 relationships with MIC scores >0.05, thus indicating weaker interactions between species. However species exhibiting the higher number of interactions included mainly parasites, nano-grazers and the rare generalist mixotrophic haptophyte (OTU00288; 100% identity to AB058358 Haptolina) that was present in all three networks. Phototrophs (Bacillariophyta) had low centralities interacting mostly between them and with nanograzers (Ciliophora) (Table S6). The top generalist but rare nanograzer OTU00409 (Spirotrichea) strongly correlated with both Syndiniales, and mixotrophs while the parasitic Blastodinium (OTU00052) interacted with mixotrophic Dinophyceae and connected to the rest of the network through OTU00056. Finally, at 2xsecchi two networks occurred with the first including only six species (mainly parasites; Syndiniales and Blastodinium) and the second including only 10 interactions between nine species and being built on the associations between parasites and algae (Chrysophyceae; Bacillariophyta) that were generalists or top generalist but not abundant (Fig. 5C; Tables 2; S7).

Discussion

Our study investigated protistan community composition (PCC) in different coastal sites and depths, from seven different gulfs in the Aegean Sea (regional Mediterranean Sea). Different depths were selected based on Secchi depth and light penetration, resulting in bottom depths for 2xSecchi in shallow sites and thus not allowing for robust conclusion regarding the effect of light penetration in protistan communities. However this was the first time that a similar study was performed to that extent (area of 72,600 km2) in the specific area (East-Mediterranean), with samples collected within a time range of 19 days, significantly minimizing the effect of temporal variation on assemblage composition, as also explained previously. Thus we believe that the results provided here will be valuable for future efforts for further disentangling protist diversity interactions in this area specifically, but also in other temperate Gulfs.

We focused on the taxonomic affiliation and ubiquitousness of different species in order to investigate interactions amongst species in different sites as well as their impact in the dynamics of protistan communities. Although ordinations using non-parametric methods provided evidence for site separation in all depths, these results were weakly supported statistically apart from the surface level (Table S3A), implying better separation in surface compared to deeper layers. However when similarities between gulfs were calculated, they were higher in surface samples compared to Secchi depth (Fig. S2), suggesting that although surface samples might be more influenced by local factors enhancing growth of specific species, differentiating community composition, they show higher dispersal. The hypothesis that epipelagic communities are more homogenous due to faster currents (Villarino et al., 2018) in surface compared to deeper layers could not be thoroughly examined in our study due to low maximum depths that in many cases corresponded to bottom water.

As suggested from previous studies relying on a synthesis of variables from hydrological, climatological and satellite data (Reygondeau et al., 2017; Ayata et al., 2018), the Aegean Sea is separated into east and west compartments fed by the Atlantic water and Black Sea outflow waters, respectively. Thus Kavala, Thermaikos and Saronikos are categorized in the west compartment, whereas sites Gera, Kalloni Kodias and Moudros are categorized in the east compartment. This partially explains the higher Shannon indices observed in the east compartment (G, K, LK, M), compared to indices observed in the West compartment (KV, T, S), (Table S2), based on the biogeography principle that larger geographical areas or water masses (Atlantic compared to Black Sea) are more species diverse as they provide the scope for more niches (Costello & Chaudhary, 2017). The potentially higher number of niches in the east compartment could also explain the higher number of top-generalists and generalists observed compared to the west part (Fig. S3). Also, the highest number of generalists explaining the differences between gulfs at surface (Table S6) could be associated with a highest number of different niches as also suggested from the distinct communities (Figs. 1; S2).

A clear top-down phytoplankton control from grazers (mixotrophs, nanograzers), was not confirmed for any depth, as shown from correlation analysis between trophic groups, for the whole community. We have to take into account though, potential bias caused by the 18S region studied. According to the PR2 primer database (v.2.0.0) (Guillou et al., 2013) our primers (V2–V3 region) probably underestimate (due to mismatches) the diversity of Excavata, Amoebozoa and TSAR (Telonemia, Stramenopiles, Alveolata, Rhizaria) compared to the respective primers for V4 and V9 regions.

However, regarding specific species that were differentially abundant between gulfs some significant interactions between algae and grazers were observed for all depths (Table S7; Fig. 5), although they were outcompeted by the strongest and more abundant interactions between algae and parasites (mainly Syndiniales). Overall parasites emerged as important contributors in protistan community composition (PCC) shaping at all depths, exhibiting strong interactions with all trophic groups (Table S7; Fig. 5).

Syndiniales are known to form parasitic relationships with other protists (dinoflagellates, ciliates, cercozoans, radiolarians) and metazoans (copepods,fish eggs), with free-living dinospores released in very high numbers following host death (Burki, Sandin & Jamy, 2021; Clarke et al., 2019), while their importance in marine trophic webs has been confirmed on a global scale after the Tara ocean (De Vargas et al., 2015) and the Malaspina-2010 expeditions (Pernice et al., 2016) that both studied 18S rRNA diversity and exhibited the prevalence of Syndiniales sequences amongst protists. It has to be noted that these studies had analyzed the V4 (Pernice et al., 2016) and the V9 (De Vargas et al., 2015) region of 18S rRNA gene, showing that Syndiniales dominance was irrespective of primers selection. However it is worth mentioning, that especially for Syndiniales high numbers might be due to their overrepresentation in DNA surveys compared to RNA surveys probably because of their high abundance in picoplankton in inactive stages with few ribosomes (Massana et al., 2015).

Although Syndinian protists parasitic lifestyle is well-known (Guillou et al., 2008) their exact ecological role remains a ‘black-box’ for marine food webs modeling. Previous studies have shown co-occurence patterns of Syndiniales with Spirotricheae and Dinophyceae exhibiting both exclusion and copresence patterns (Anderson & Harvey, 2020). Other studies, working on high-resolution time-series in a productive coastal pond (Sehein et al., 2022), indicated seasonal shifts in protist populations, exhibiting elevated concentrations of free-living parasitic Syndiniales along with their infected dinoflagellate hosts during high productivity periods, while others (Long et al., 2021) have proved the existence of Dinophyceae ‘weapons’ against Syndinians contributing in avoiding the total collapse of Dinophyceae blooms. A recent seasonal study in a meso-eutrophic coastal sytem, focusing on Syndiniales Group II has shown that Syndinians peaked in summer along with Prorocentrum minutum, while in autumn had diverse host (Christaki, Skouroliakou & Jardillier, 2023).

These previous findings corroborate with the strong interactions detected at Secchi depth between Dinophyceae and Syndiniales (Fig. 5B) as well as with the significant interactions between mixotrophs and parasites in total (Table S4) and with the co-presence of abundant populations of Group II Syndiniales with their potential hosts, such as Gyrodinium, Hetrocapsa, Scrippsiella and Prorocentrum (Coats et al., 1996; Chambouvet et al., 2008; Salomon et al., 2009; Christaki, Skouroliakou & Jardillier, 2023). These findings and interactions potentially support previous studies suggesting the importance of parasites in controlling mortality in marine systems, influencing Dinophyceae bloom dynamics and species succession, (Chambouvet et al., 2008; Skovgaard et al., 2005; Jephcott et al., 2016).

However Syndiniales associations with autotrophs are poorly studied. Investigations performed for Bacillariophyta and Mamielophyceae mostly indicated exclusion (Anderson & Harvey, 2020), with the authors suggesting resource competition (either direct or indirect) between taxa, as nutrients such as silicates were important in explaining patterns in Syndiniales composition. Another study (Sassenhagen et al., 2020) also indicated direct and indirect interactions between diatoms and Syndinians by studying the infection of diatoms by Syndiniales also exhibiting host specificity despite their large host range. Thus, although the nature of the strong interactions observed in surface and 2XSechi (Syndiniales vs Chlorophyta, Bacillariohyta, Chrysophyceae) cannot be easily clarified, direct (resources competition) or indirect (effect on phytoplankton grazers) interactions could be suggested.

Blastodinium is a well-known copepod parasite (Skovgaard, 2005; Alves de Souza et al., 2011) and interactions detected in our study with Dinophyceae probably do not reveal trophic relationship but co-occurrence inside the host (i.e., copepod feed). The conspicuous role of the generalist but very rare OTU00288 (Prymnesiales) interacting with members from different trophic groups at each depth further confirmed the potential versatility of haptopyte species exhibiting different lifestyles swiping from autotrophy to heterotrophy, but also being able for toxic blooms formation (Johannessen et al., 2015) However interactions mainly involving exclusions have been detected between Syndiniales and Prymnesiophyceae (Anderson & Harvey, 2020).

Previous studies have suggested that rare microbial community members might assist in community stabilization due to their ability to rapidly respond to environmental changes (Shade et al., 2014). This was further confirmed from our results since most of the species exhibiting significant interactions were generalists but not abundant members of the total population highlighting the importance of the rare biosphere.

Conclusions

We studied the spatial structure of the entire protistan community in seven different gulfs and three different depths in a regional Mediterranean Sea. Our aim was to define taxa that are important for explaining protistan community differences observed across the different gulfs studied as well as their trophic interactions. Although we hypothesized that clear interactions between phytoplankton species and their grazers would be important for explaining these differences, we found that parasites (expecially Syndinians) were ‘key species’ in all depths across different gulfs. Distinct associations between parasites (mainly Syndiniales), with phototrophs and and mixotrophs appeared as important, for spatial differentiation, in surface/2xSecchi and Secchi depth respectively, suggesting novel direct or indirect trophic relationships. Nevertheless, due to the high number of potential hosts that could vary from algae to metazoans (crustaceans, fish), it was not feasible to disentangle whether parasites interactions with the other protists were direct or indirect. Thus, the dynamics of this group that appeared as an important player in shaping protist variation could be influenced by factors (such as fish stock) far beyond the traditional factors measured in traditional ecological studies. Although our findings agreed with previous studies suggesting the importance of parasites in controlling Dinophyceae bloom dynamics and species succession and mortality in marine systems, influencing , this was the first time, to our knowledge, that phototrophs-Syndinians associations emerged as an important factor for shaping protist community composition in a coastal environment, presenting thus a novel perspective regarding parasites associations.

Supplemental Information

Supplemental Information 1 Supplementary tables

Click here for additional data file.

Supplemental Information 2 Supplementary figures

Click here for additional data file.

Additional Information and Declarations

Competing Interests

Author Contributions

Data Availability

Konstantinos Kormas is an Academic Editor for PeerJ.

Alexandra Meziti performed the experiments, analyzed the data, prepared figures and/or tables, authored or reviewed drafts of the article, and approved the final draft.

Evangelia Smeti performed the experiments, prepared figures and/or tables, authored or reviewed drafts of the article, and approved the final draft.

Daniil Daniilides conceived and designed the experiments, performed the experiments, authored or reviewed drafts of the article, and approved the final draft.

Sofie Spatharis conceived and designed the experiments, performed the experiments, prepared figures and/or tables, authored or reviewed drafts of the article, and approved the final draft.

George Tsirtsis conceived and designed the experiments, authored or reviewed drafts of the article, and approved the final draft.

Konstantinos A. Kormas analyzed the data, authored or reviewed drafts of the article, and approved the final draft.

The following information was supplied regarding data availability:

The 18S rRNA amplicon sequences are available at GenBank: PRJNA515026.

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
