# Peer review of "Increased contribution of parasites in microbial eukaryotic communities of different Aegean Sea coastal systems"

_PeerJ, doi:10.7717/peerj.16655_

## Round 0.1 · original submission · Major Revisions

The manuscript deals with the dynamics of protist communities in the Aegean Sea. Two reviewers have given their comments and I urge the authors to follow those questions carefully to respond. English needs to be checked according to a reviewer, kindly follow this.

**Language Note:** The Academic Editor has identified that the English language must be improved. PeerJ can provide language editing services - please contact us at [email protected] for pricing (be sure to provide your manuscript number and title). Alternatively, you should make your own arrangements to improve the language quality and provide details in your response letter. – PeerJ Staff

·

Basic reporting

• English needs to be checked as some sentences are not clear (e.g., L98-100: do you mean “remains rather unclear”?; L114-118: the sentence is too long and needs to be split for better understanding; L124-126: I not sure I understood this sentence; L134: replace target by objective; etc).
• I think the title should be rephrased for clarity, something like “Increased parasites contribution in microbial eukaryotic community of the Aegean coastal water”.
• NGS (next gen sequencing; L 101) is an old terminology and should be replaced by HTS (high-throughput sequencing)
• Also there are many typos. Here are few of them: L211: “Similariites”, L 286 & 353 & 358 “MICE”?, L365 “Syndiniales ,and mixotrophs while”, L 366 with.mixotrophic, L418 “cilates”, L426 “because of to”, L435 “dinoflaggelate”, L439 “Prorocenrum”
• There is many supplemental information, some should most likely be included in the main text (e.g. Fig S2 cited six times)
• There is not figure 6 (L 442) and Table 3 (L334, 337)
• Please homogenize OTU notation (OTU59 or Otu00059)

Experimental design

• Can the authors better explain the sampling strategy? There is seven coastal site (not nine as mentioned in Table 1) and five stations per site? and three depths per station, so 105 samples (or 135 if really nine sites). So, what are the 112 samples mentioned? the map Fig S1 can be useful in the main text to understand distance between sites. Kodias or Kontias ?
• Can the authors justify the use of the v2-v3 primer? Most of the works on microbial eukaryotic community has been done using the V9 primer (de Vargas et al 2015) or the V4 primer (Stoeck et al 2010, Pernice et al 2016). How do your primer set compare to the others?
• The bioinformatics is not really clear and not common for eukaryotic work. Please see Pernice et al 2016, or Christaki et al 2023 for common practice. L193 you mention SILVA but I am not sure for what and then L202 you mentioned PR2 (not PR4). Generally, the sample need to have the same depth to be compared. You calculated Bray-Curtis similarities using Hellinger transformed abundance, so you actually using Manhattan distance. To use Bray-Curtis similarities index you should use untransformed data with the risk of uneven sampling depth or rarefy your samples before calculation. The same should be applied to compare alpha-diversity.

Validity of the findings

• More details about the environment are needed to conclude that each sample or gulf represented a different niche (L225-226 and L329-331). Levins’ niche index is interesting here but needs to be used on contrasted environment (the R of the index means differing environments). A top generalist needs to be observed in more than 15 samples and so can be present in only 1 site? Did you remove the replicate samples before performing this analysis? I think some work/analysis showing the difference between samples, depth, stations, sites are therefore needed (e.g., PCA), especially given the communities are grouping/clustering by site for each depth. What about the depth? Are the communities sampled at different depths for a station, for a site, and across sites similar?
• Some clarifications are needed about OTU classification as abundant or rare. Are you considering 1% in the whole dataset or 1% in a sample? Are you considering the raw data or the transformed data? If it is 1% in the whole dataset, how can be relative abundance of the 25 abundant OTUs less than 5%? IF it is 1% in a sample, how you classify an OUT with 2% in a sample and 0.1% in another?
• Can you explain the difference between nanograzers and nanoheterotrophs? Why are pico- and micro-heterotrophs missing in figure 6?
• L405-407: a better analysis of environmental parameters, as suggested above, can help with this aspect.
• I do not understand your network analysis. The idea is generally to summarize the interaction in an ecosystem, but you decided to only consider OTUs differentially abundant across sites. Main players such as OTUs abundant at all sites, or without abundance variations across sites were excluded from the analysis. This aspect should be more discussed, especially the lack of (L481) “clear interactions between phytoplankton species and their grazers”.
• L479-480: “explaining microbial network differences observed across the different gulfs studied”. You performed network per depth but not per site, so how can you have studied the difference between gulfs?
• Some results are presented (e.g., L276-277) but not discuss.

Reviewer 2 ·

Basic reporting

no comment

Experimental design

no comment

Validity of the findings

no comment

Additional comments

See attached PDF

Annotated reviews are not available for download in order to protect the identity of reviewers who chose to remain anonymous.

---

## Round 0.2 · Minor Revisions

The revised manuscript has been improvised well, however it is still found with some minor mistakes in the typo and in writing as per the reviewers' comments. Reviewer advised the authors to focus on the limitation of your methodology in the discussion part. Please follow the two reviewers comments carefully and resubmit it.

·

Basic reporting

Still a few typos. Please check the track change documents attached.

Experimental design

The primers were selected in 2016 since the same samples were used for phytoplankton community analysis in the same area (Spatharis et al., 2019). The use of these primers by Genitsaris et al. 2015, successfully detecting the majority of eukaryotic groups in the English Channel further supported our choice.
=> I still think some discussion should be included in the manuscript, especially to broader the conclusions by comparing with other studies.

Thank you for this comment. SILVA was used for the alignment and PR2 for the taxonomic classification that is the common practice when using mothur (alignment and then taxonomy). Christaki et al. 2023 have used completely different methods since they analyzed ASVs, while in Pernice et al. 2016 they performed pyrosequencing and also used SILVA and PR2.
We agree with your comment regarding Bray-Curtis and Hellinger, thus we recalculated all bray-curtis similarities with non-transformed data. (l.219-220/l.277-282)
=> While I agree that Pernice et al 2016 used a similar approach, the method is at least 7 years old and things are changing rapidly in bioinformatics and in regard of knowledge on protists. As mentioned by the reviewer 2, a clustering approach at 97% does not look appropriate for protists, because some clades (e.g., dinoflagellates) will show little variation in their SSU sequences and therefore you will collapse several taxa in one OTU. Now, with your goal of looking at difference in taxa composition between habitats, you ran the risk of losing signal using this approach and you should acknowledge this possibility in the discussion.

Other comments on the attached rebuttal with track changes.

Validity of the findings

no comment

Reviewer 2 ·

Basic reporting

Considering the improvement of the manuscript through changes in the MS as asked and through detailed and adequate answers in my comments, this work could be published as it is. I only have two remarks:
- line 112 please add a comma after 'Caruso et al.' and remove '., ' after the parenthesis.
-Also, I could not find the explanation of the MIC in lines 255-260. Could you please add 'MIC captures associations between data and provides a score that represents the strength of the relationship between data pairs.'

Experimental design

no comment

Validity of the findings

no comment

Additional comments

no comment

---

## Round 0.3 · accepted · Accept

Authors have sent the satisfied revision, I accept your manuscript for publication.